# Supramolecular metallic foams with ultrahigh specific strength and sustainable recyclability

Xin Yang[1], Xin Huang[1], Xiaoyan Qiu[1], Quanquan Guo [2] & Xinxing Zhang [1]✉

Porous materials with ultrahigh specific strength are highly desirable for aerospace, automotive and construction applications. However, because of the harsh processing of metal foams and intrinsic low strength of polymer foams, both are difficult to meet the demand for scalable development of structural foams. Herein, we present a supramolecular metallic foam (SMF) enabled by core-shell nanostructured liquid metals connected with high-density metal-ligand coordination and hydrogen bonding interactions, which maintain fluid to avoid stress concentration during foam processing at subzero temperatures. The resulted SMFs exhibit ultrahigh specific strength of 489.68 kN m kg$^{-1}$ (about 5 times and 56 times higher than aluminum foams and polyurethane foams) and specific modulus of 281.23 kN m kg$^{-1}$ to withstand the repeated loading of a car, overturning the previous understanding of the difficulty to achieve ultrahigh mechanical properties in traditional polymeric or organic foams. More importantly, end-of-life SMFs can be reprocessed into value-added products (e.g., fibers and films) by facile water reprocessing due to the high-density interfacial supramolecular bonding. We envisage this work will not only pave the way for porous structural materials design but also show the sustainable solution to plastic environmental risks.

Advanced structural materials with ultrahigh strength and lightweight are highly desirable for a wide variety of technological applications, including aerospace[1,2], automotive[3,4], and construction industries[5,6]. Constructing porous architectures is a proven approach to realize weight reduction but usually lead to the decrease of mechanical properties, making it difficult to meet structural material requirements[7,8]. To date, it is still a great challenge to achieve ultrahigh mechanical properties of lightweight porous materials.

Design of foams, by the methods of introducing bubbles[9], vapor deposition[10], 3D printing[11,12], freeze casting[13–15], and so on, is recognized as one of the most effective approaches to achieve lightweight structural materials[16]. Pham et al. developed architected porous materials that are robust and damage-tolerant enabled by mimicking the microscale structure of crystalline materials[17]. Yang et al. demonstrated light and stable nanoporous aluminum (Al) via galvanic replacement reaction, which is stronger than conventional foams of similar density consisting of pure Al or Al-based composites[18]. Despite the encouraging early works, the conflict between lightweighting and outstanding mechanical properties of metallic foams remains irreconcilable due to the high density of metals themselves. In addition, the manufacturing processes of metal foams are energy-intensive and resource-wasting, and frequently require harsh conditions including temperature control above 600 °C and high-pressure gas, restricting their further development[19].

Combination of the intrinsic merits of metals and polymers to develop composite foams to achieve lightweight is a cutting-edge

[1]State Key Laboratory of Polymer Materials Engineering, Polymer Research Institute, Sichuan University, Chengdu 610065, China. [2]Max Planck Institute of Microstructure Physics, Halle (Saale), 06120, Germany. ✉e-mail: xxzwwh@scu.edu.cn

research direction[20–22]. However, since the modulus mismatch as well as the lack of interfacial interaction between polymers and hard metals result in deterioration of mechanical properties (compressive strength usually <1 MPa)[23,24]. More importantly, these types of foams are also difficult to recycle or reprocess after damaged. The disposal or landfill of them at the end of life can easily cause plastic environmental issues[25–27].

With these concerns in mind, herein, we propose a supramolecular metallic foam (SMF) enabled by core-shell nanostructured liquid metals (LMs), which connected with high-density supramolecular interactions of metal-ligand coordination and hydrogen bonding interactions, to achieve ultrahigh specific strength and sustainable recyclability. Nanoscale LMs particles (LMNPs) formed based on ultrasound-induced acoustic cavitation exhibit oxidized skins with high specific surface area to form metal-ligand coordination with polyacrylic acid (PAA) for constructing cross-linked network. Nanosizing also enables interface energy to be elevated to increase the Gibbs free energy of the system thereby inhibiting the crystallization of LMs, allowing the composites to be shaped at subzero temperatures. Hence, the resulted SMFs can exhibit ultrahigh mechanical properties and low densities, which is superior to Al alloy foams. In addition, they can also be easily reprocessed to other value-added products once the foams have reached the end of service life, such as fibers and films. This strategy might not only open new avenue towards the resolve of conflict between lightweight and high strength of materials, but also the sustainable applications serving a variety of strategic areas.

## Results

### Materials design of SMFs

To overcome the challenge of fabricating lightweight foams with high specific strength, the strategy of supramolecular core-shell nanostructure is proposed as shown in Fig. 1a. Compared to traditional hard metals, LMs have covalent properties that lower the melting point[28]. Among them, LMs of Eutectic gallium-indium (EGaIn) have gained widespread attention owing to their low melting point (15.7 °C) and low bulk viscosity ($1.99 \times 10^{-3}$ Pa·s)[29–32], which demonstrate metallic core and soft flexible shape.

In order to fabricate SMFs, first of all, acoustic cavitation induced by ultrasonication can cause elongation and breaking of bulk LMs and create LMNPs because of the large surface tension. LMs develops a near-instant gallium oxide layer consisted primarily of $Ga_2O_3$ under oxygen environment[33]. When LMs and acrylic acid are sonicated simultaneously, the oxide skins on the surface of LMs are continuously destroyed by acrylic acid (Hydrogen ions ionized in acrylic acid can react with $Ga_2O_3$ to form $Ga^{3+}$ and water[34]) and grow new oxide layers. Taking advantage of the dynamic cross-linked interface of LMNPs, acrylic acid is polarized under the ionic potential of them to cover the surface of them and then polymerized under ultrasonication. The formation of core-shell nanostructure based on metal-ligand condonation between LMNPs and acrylic acid, shown in Fig. 1b and Supplementary Fig. 1, reduces the surface tension of LMs and contribute to the dispersion and stabilization of LMNPs. Zeta potential, defined as the degree of electrostatic repulsion between charged particles in a

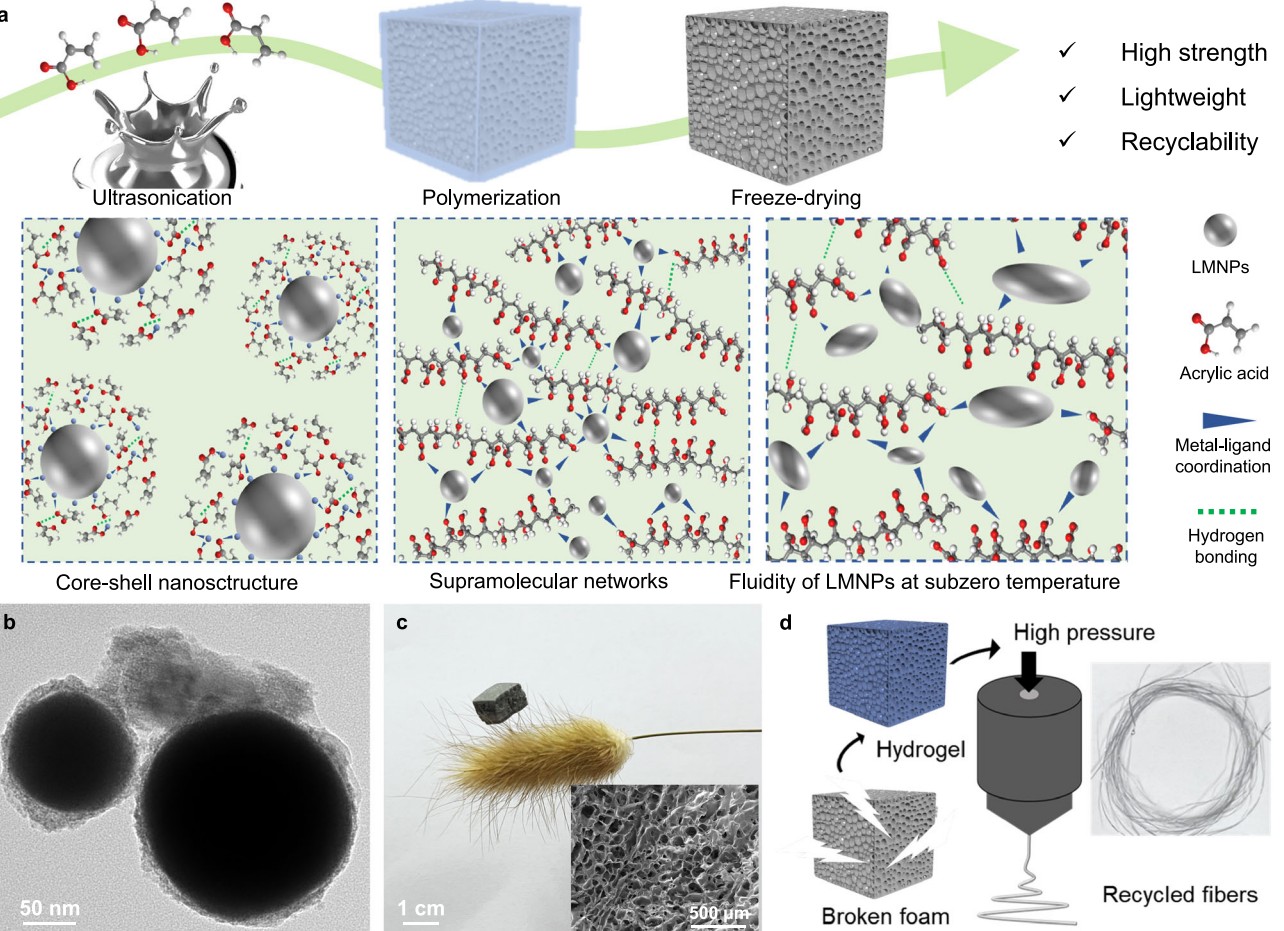

**Fig. 1 | Chemical structure, design concept, and recyclability of SMFs.**
**a** Schematic illustration of the fabrication process and supramolecular networks of SMFs. **b** TEM micrograph of LMNPs after ultrasonication with acrylic acid. **c** Digital photograph of SMFs supported by bristlegrass, and the insert is cross-section SEM image of SMF. **d** Schematic illustration of end-of-life SMFs reprocessed into fibers.

dispersion[35], was used to analyze the assembly mechanism of core-shell nanostructure (Supplementary Fig. 2). The change in zeta potential with time proves that acrylic acid monomer progressively alter the surface charges during ultrasonication. The repulsive force between nanoparticles increases due to the enhanced negative charge carried. At this time, both the oxidized skin of LMNPs and the free $Ga^{3+}$ (Supplementary Fig. 3) in the vicinity will form metal-ligand coordination with carboxyl groups of acrylic acid thus forming nanostructured supramolecular networks. On the other hand, the rapid polymerization of acrylic acid anchors LMNPs to prevent settling. Therefore, LMNPs with high specific surface area have a huge number of dynamic cross-linking sites to form metal-ligand coordination with PAA. Different from the conventional chemical cross-linking, the construction of such supramolecular cross-linking networks based on LMNPs restrict the movement of PAA macromolecular chains to a certain extent yet dissipate energy through deformation at the same time, which contributes significantly to the enhancement of the mechanical properties of the metallic foams[36]. Furthermore, nanoscale confinement keeps the surface atom of LMNPs in an unstable state, which makes their surface lattice vibration larger to decrease the melting temperature of LMs. A small peak at −26 °C is attributed to the phase transition peak of a very small number of large-sized LMNPs. Hence, obvious phase change won't occur in LMs during the freeze-drying process (Supplementary Fig. 4), which aids in the load transfer of the process and prevents hole collapse due to stress concentrations (Supplementary Fig. 5). The coordination bonds between Ga (III) and carboxylate have relatively higher kinetic activity for dissociation and recombination. Such coordination bonds make it difficult to form periodic structures such as crystals (Supplementary Fig. 6), and the LMNPs were uniformly distributed in the foam skeleton without agglomeration (Supplementary Figs. 7–9). Therefore, the produced foams have low densities, which can be easily supported by bristlegrass without any collapse of the bristlegrass itself (Fig. 1c). The SMFs show significant performance compared to polyurethane (PU) foam and Al alloy foam[37,38] (Supplementary Fig. 10, Supplementary Table 1).

Considering the construction of supramolecular networks, it was anticipated that SMFs would display emerging features that are rarely processed by traditional foams. In particular, conventional plastic foams cannot be recycled or reprocessed in mild condition, which usually result in serious white pollution[39,40]. Nevertheless, taking advantage of the water sensitivity of supramolecular networks, the metallic foams can be easily aqueous-reprocessed into high-value fibers, which is shown in Fig. 1d, to avoid causing serious environmental problems. The elaborate design of supramolecular networks between metals and macromolecular chains provides not only satisfactory mechanical properties of lightweight foams but also relatively sustainable recyclability.

## Supramolecular core-shell nanosctructure

While LMs and acrylic acid are sonicated at the same time, the oxide skins on the surface of LMs are continuously destroyed by acrylic acid and new one grows. At this time, both the oxidized skin of LMNPs and the free $Ga^{3+}$ in the vicinity will form metal-ligand coordination with carboxyl groups on the macromolecular chains thus forming nanostructured supramolecular networks, which is shown in Fig. 2a. Laser confocal Raman microspectroscopy was used to demonstrate the presence of gallium-carboxyl complexation in metallic foams. As shown in Fig. 2b, the C = O stretching vibrational peak shift from 1668 to 1719 cm$^{-1}$, which proves that the formation of interactions between $Ga^{3+}$ and carboxyl groups[41]. 2D Raman mapping was developed by measuring the intensity of 1719 cm$^{-1}$ on the surface (20 μm * 20 μm) of SMF to intuitively illustrate the distribution of gallium-carboxyl complexation. As shown in Fig. 2c, 2D Raman mapping was developed by measuring the intensity of the peak at 1719 cm$^{-1}$ in different locations

on the surface of the material. The color bar on the right shows the relationship between color and the associated value. Specifically, the strong color contrast between different areas indicates that uneven distribution of carboxyl groups complexed by gallium, which is attributed to the fact that the complexation is related to the distribution of LMNPs. Furthermore, the high peak intensity proves the high-density metal-ligand interactions of SMF compared to that of PAA foam (Supplementary Fig. 11). The above findings further demonstrate the construction of supramolecular networks based on LMNPs as noncovalent cross-linking sites.

In order to further understand the interaction mechanism of SMFs, temperature-dependent Fourier transform infrared (FTIR) spectroscopy was performed to characterize the gallium-carboxyl complexation and hydrogen bonding interactions. As illustrated in Supplementary Fig. 12a, the peak at 1708 cm$^{-1}$ is assigned to the −C=O groups of PAA. The spectral intensity of 1766 cm$^{-1}$ which is assigned go the stretching vibration of "free" −C=O groups gradually increase during the heating process[42,43]. In addition, as shown in Supplementary Fig. 12b, the peak centered at 3251 cm$^{-1}$ is belong to the stretching vibration band of -OH, whose intensity decreases obviously and shift to 3313 cm$^{-1}$ upon heating[44]. These results prove that multiple hydrogen bonds exist in the resultant metallic foams. Furthermore, two-dimensional correlation spectra were performed to investigate the dissociation order of the interactions. Based on the sign of the correlated peaks in the synchronous and asynchronous spectra, the order of spectral intensity changes for a given wavenumber can be easily determined according to Noda's rule. The generalized 2D correlation FTIR spectra in the region of 1850–1650 cm$^{-1}$ and 3550–3000 cm$^{-1}$ are shown in Fig. 2d, e and Supplementary Fig. 13. According to the rule, the disassociation of "bonded" −C=O is before formation of "free" −C=O, which is ascribed to that the "bonded" −C=O tends to form metal coordination with $Ga^{3+}$ while disassociation[45]. Furthermore, perturbation−correlation moving window two-dimensional (PCMW2D) correlation spectroscopy was used to investigate the precise dissociate temperature of hydrogen bonds[46]. As illustrated in Fig. 2f, hydrogen bonds are highly active over a wide temperature range[47]. The results above show that multiple noncovalent interactions in the systems are highly active, which lays a foundation for the energy dissipation under the external force of the metallic foams.

Due to the frozen motion of PAA macromolecular chain segments at room temperature, the activation energy and multiplicity of secondary relaxation, including the small motor units such as branching units and end groups, are considered to be important factors affecting the mechanical properties of SMFs. To study the factors influencing molecular dynamics by nanostructured supramolecular networks of obtained foams, temperature-dependent dielectric loss spectra (Fig. 2g and Supplementary Fig. 14) were conducted. As shown in Supplementary Fig. 15, the dielectric loss spectra of PAA/EGaIn as a function of frequency show three relaxation processes by analyses of Havriliak−Negami (H−N) function[48], including β-relaxation (the local fluctuations of end groups including carboxyl-terminal), γ-relaxation (the crank motion of free carboxyl groups) and δ-relaxation (the twisting and swinging motion). The average relaxation times of β, γ, and δ-relaxation extracted from the H-N equation can be plotted as function of temperatures, which is shown in Fig. 2h. It can be observed directly that the kinetic rate of δ and γ-relaxation are faster than that of β-relaxation, and all of them can take place at room temperature or even at 0 °C. These results demonstrated that the end groups and functional unites of PAA/EGaIn are mobile in glassy state, which contributes to increased energy dissipation while maintaining mechanical properties. Afterward, the Arrhenius function was utilized to describe the average relaxation time as a function of temperature, and activation energies ($E_a$) for the motions of different moieties were obtained[49,50] (Fig. 2i). The $E_a$ values of three relaxation for PAA/EGaIn and PAA are quite low (all below 50 kJ/mol). This result demonstrates

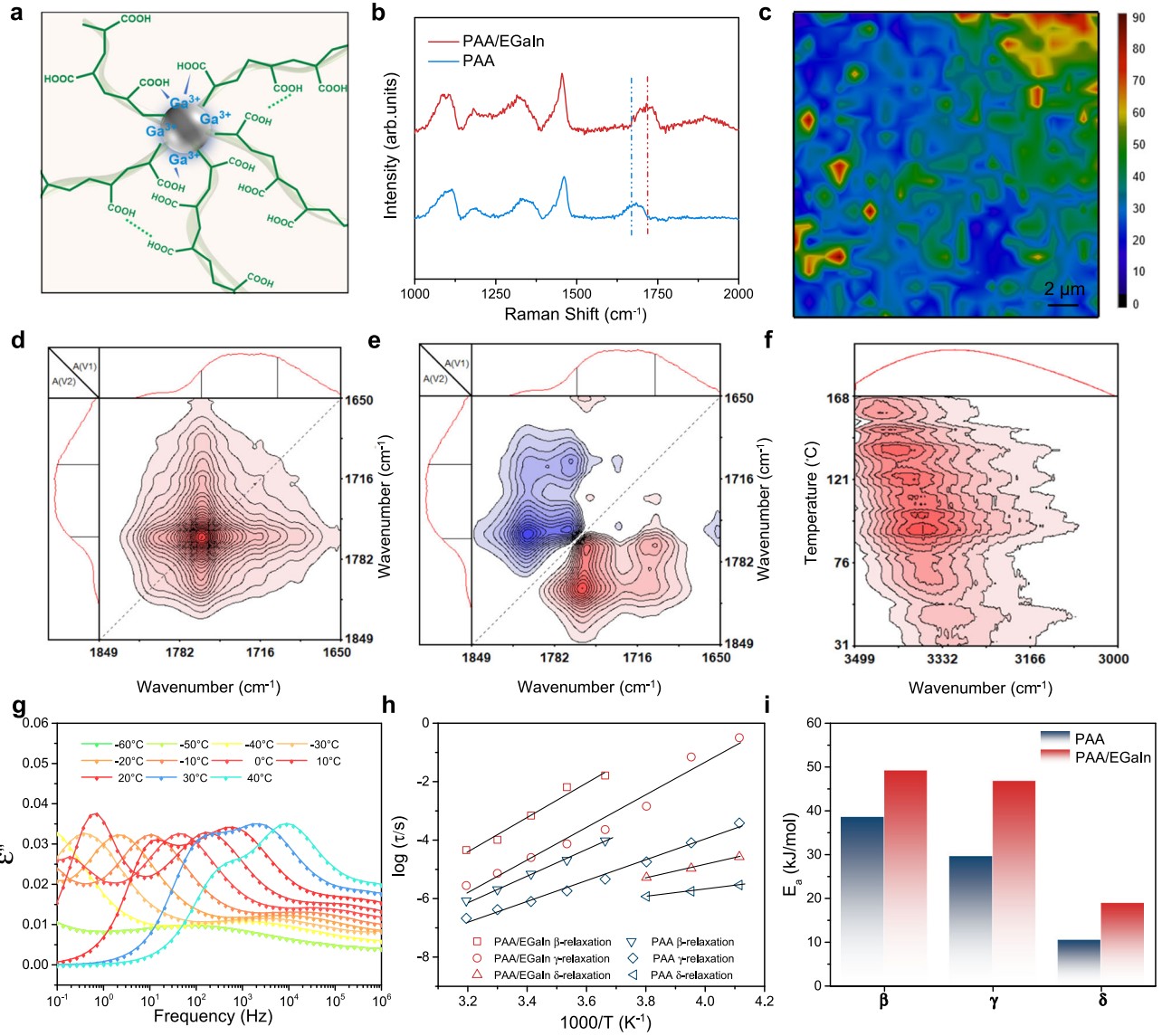

**Fig. 2 | Characterization of supramolecular networks. a** Schematic illustration of the supramolecular networks around LMNPs. **b** Laser confocal Raman spectra of SMF and PAA foam. **c** 2D Raman intensity surface mapping at intensity of 1719 cm⁻¹ of obtained SMF. Synchronous (**d**) and asynchronous (**e**) generalized 2D correlation spectra from 1850 to 1650 cm⁻¹ of SMF during heating from 20 to 180 °C. **f** PCMW2D synchronous spectra from 3500 to 3000 cm⁻¹ of SMF during heating from 20 to 180 °C. **g** Dielectric loss ε″ as a function of frequency for PAA/EGaIn from −40 to 40 °C. **h** Average relaxation time as a function of temperature for β, γ, and δ-relaxation of PAA/EGaIn and PAA. **i** Activation energies ($E_a$) of PAA/EGaIn and PAA β, γ, and δ-relaxation.

the reason why these relaxation modes can easily occur in the glassy state. However, the $E_a$ of three relaxation modes for PAA/EGaIn are higher than that of neat PAA, which is attributed to the fact that the formation of supramolecular networks restricts the motion of the end groups and functional unites. The construction of such robust yet high active dynamic networks is important for achieving high mechanical properties of metallic foams.

## Mechanical properties

To evaluate the practicability of SMFs in lightweight structure materials, mechanical performance was studied. The load-bearing capacity of the foam skeleton is significantly improved attributed to the construction of nanostructured framework and supramolecular networks, which was proven by the nanomechanical mapping of atomic force microscope (AFM)[51]. Figure 3a shows the distribution of surface modulus of PAA foam and SMFs, respectively. The average Young's modulus for the surface layer of PAA foam is 1.12 GPa, which is much lower than that of 8.61 GPa for SMFs.

Moreover, the compressive stress-strain curves of SMFs with different ratios of PAA and EGaIn were shown in Fig. 3b, which exhibit three distinct deformation stages (SEM images of different stages shown in Supplementary Fig. 16). During the first stage, a linearly increasing initial Hookean regions was found at a small strain (about 5%–10%), indicating definite elasticity of the metallic foams. The pore morphology of foams doesn't change noticeably. Subsequently, plateau stages of the strain less than about 60% with slowly increased stresses were observed after yielding, which can be attributed to the deformation of LMNPs and partial squeezing of the porous structures. Partial squeezing of the porous structures was found. Ultimately, as the porous structures were further compressed, the stresses increased dramatically. This is because most of the holes are compressed and the high strength provided by the skeleton. These observations demonstrate that the skeleton of SMFs has superior rigidity along with excellent deformability without brittle damage, realizing the combination of high strength and toughness. PAA foam crosslinked by Ga³⁺ exhibits brittleness, which is easily damaged

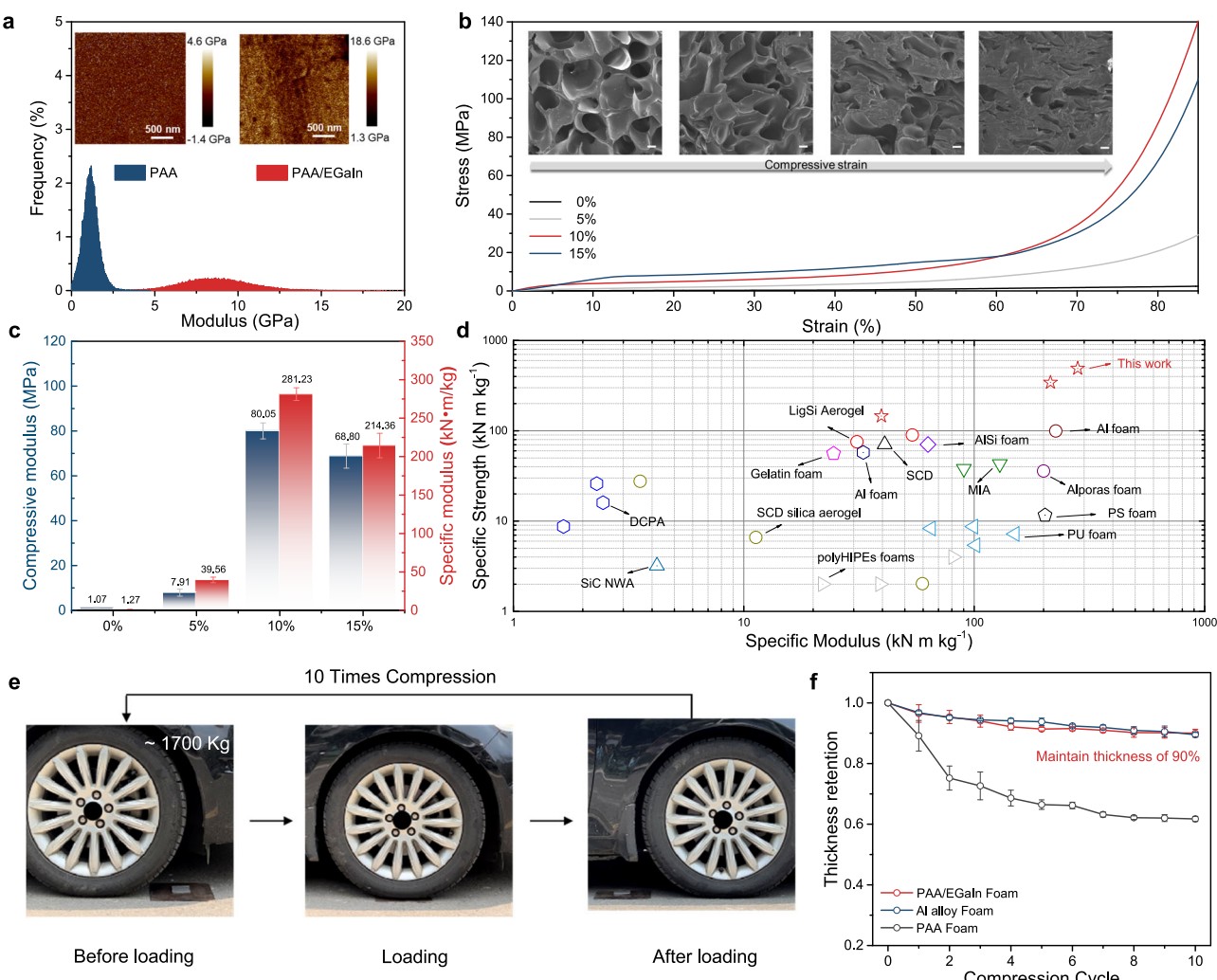

**Fig. 3 | Mechanical properties. a** Modulus images and the corresponding distributions of modulus for PAA foam and SMF obtained from AFM. **b** Compressive stress-strain curves of SMFs with different content of LMNPs under a strain of 85% with a deformation of 10 mm min⁻¹, and the insert is cross-section SEM image of SMF (10 wt% EGaIn) under the compressive strain of 0%, 5%, 50% and 85% (Scale bar: 20 μm). **c** Compressive modulus and specific compressive modulus of SMFs with different content of LMNPs calculated from their corresponding stress-strain curves; three measurements were conducted for each data point with the error bars corresponding to the standard deviation. **d** Comparison of the specific strength and specific modulus of SMFs and other reported foams. **e** Digital photographs showing the car loading test of SMF of 10 cycles. **f** The thickness retention of SMF, Al alloy foam, and PAA foam after the loading of the car for 10 cycles; three measurements were conducted for each data point with the error bars corresponding to the standard deviation.

during compression (Supplementary Fig. 17). Moreover, finite element analysis has been conducted to study how soft cross-linking points change the strength of the system. As shown in Supplementary Figs. 18, 19, large deformation of LMNPs reduces the load at the interface between LMNPs and the polymer matrix. On the contrary, higher stress at the surface between the polymer matrix and rigid nanoparticles can easily damage or destroy the materials. This result suggests that flowability and large deformation of LMNPs can transfer and dissipate energy under stress, thus avoiding damage of foams due to stress concentrations. The compressive modulus and specific compressive modulus (defined as the ratio of compressive modulus to density, an important parameter reflecting the mechanical strength of foam) of SMFs were described in Fig. 3c. It's quite clear to visualize that with the introduction of LMNPs, the modulus of the foam increases tremendously. The SMFs with 10% LMNPs content shows the highest specific compressive modulus of 281.23 kN·m·kg⁻¹, which is 221.44 times more than that of PAA foam. When the content of LMNPs is further increased, the inherent high density of LMNPs results in a decrease in the specific modulus of foam. These observations indicate that the supramolecular linking of metals and

polymers play important roles in maintaining a strong skeleton of the foam. In addition, compared to those of traditional polymer or inorganics-based foams, and even to metal foams such as aluminum, SMFs has ultrahigh specific compressive strength and modulus to meet the requirements of load-bearing scenarios (Fig. 3d and Supplementary Table 2). Moreover, although the mechanical properties of SMF were decreased with the increased humidity as shown in Supplementary Fig. 20. Compressive modulus of SMF at humidity of 90% is still about 5 times higher than PU foam.

In order to evaluate the load-bearing performance of SMFs in practical utilization situations, a four-wheel car with total weight of about 1700 kg was driven and held on the top of SMF, Al alloy foam and PAA foam placed on the copper sheet to avoid stress concentrations due to uneven ground surfaces. Subsequently, the car repeatedly compacted the foam 10 times as shown in Fig. 3e, and no fracture or crushing of SMF was observed. The thickness variations of SMF, Al alloy, and PAA foams are shown in Fig. 3f. After repeated loading of the car, SMF maintains the thickness of 90%, which is comparable to Al alloy foam. However, the skeleton of the PAA foam has collapsed with a deformation of about 40%.

 

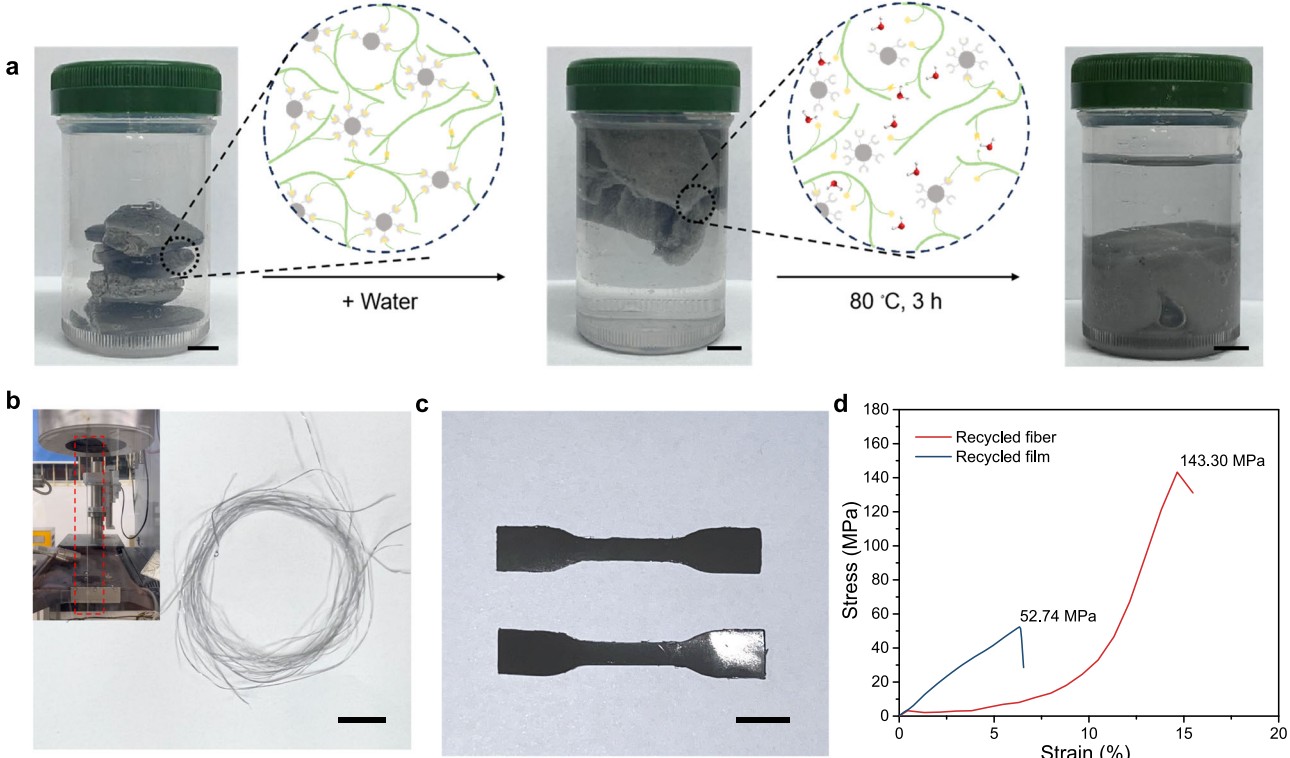

**Fig. 4 | Characterization of recyclability. a** The recycle process of end-of-life SMFs (Scale bar: 1 cm). **b** The digital photograph of reprocessed fibers by SMFs, and the insert is photograph of the spinning process (Scale bar: 1 cm). **c** The digital photograph of reprocessed films by SMFs (Scale bar: 1 cm). **d** The tensile stress-strain curves of recycled fiber and film.

## Aqueous-reprocessability for sustainable recycling

Irreversible covalent cross-linking of macromolecular chains makes recovery of polymer foams challenging as well and narrows their end-of-life options to incineration or landfill[52]. The development of sustainable recycling strategies for foams is therefore essential to manage the significant quantities of waste. In consideration of the abundant supramolecular networks in SMFs, it is expected that these foams exhibit emerging features which are otherwise hardly possessed by traditional foams. It's important to note that dynamic networks can realize reversible dissociation and reconfiguration in response to external stimuli, which is beneficial for foam recovery under mild conditions.

The recycling process of damaged SMFs is illustrated in Fig. 4a. When the foams were treated with water, the dissociation of hydrogen bonding of metal coordination bonds reduced the restriction of PAA chain movement, and heating accelerates this process. The movement of the activated macromolecular chains allows them to diffuse into each other, which leads to softening of the foams into hydrogels. The softened hydrogel can be molded into a variety of value-added products. Fibers and films were presented as two examples to demonstrate that the strategy can convert end-of-life foams into value-added products under mild processing conditions. As shown in Fig. 4b, the gel was extruded by high-pressure capillary rheometer at room temperature under high-pressure conditions, and fibers were obtained by drying at ambient conditions (Fig. 4c). On the other hand, the films were obtained by hot-pressed of gels at 60 °C and 10 MPa for 30 min. In order to verify the serviceability of recycled products, mechanical performance was measured. As shown in Fig. 4d, the tensile strength of recycled fiber and film are 143.30 MPa and 52.74 MPa, respectively. Overall, the strategy shows the potential for a cradle-to-cradle life-cycles as the foams can be reprocessed into fibers and films by facile aqueous reprocessing.

## Discussion

Nanoscale LMNPs exhibit high specific surface area as physical cross-linking sites, whose oxidized skins can form abundant metal-ligand coordination to provide high skeleton strength. At the same time, the flowability of LMNPs helps transfer and dissipate energy under stress, thus avoiding damage of foams due to stress concentrations. Furthermore, LMNPs also contribute to the foam-shaping process. Nanosizing increases the Gibbs free energy of the system thereby inhibiting the crystallization of LMs during freeze-drying, which aids in the load transfer of the process and prevents hole collapse. Based on the synergistic interactions of metal-ligand coordination and hydrogen bonding, the movement of PAA macromolecular chains is restricted to a certain extent (modulus of foam skeleton increase from 1.12 GPa to 8.61 GPa) yet dissipate energy through deformation at the same time, which contributes significantly to the enhancement of the mechanical properties of the foams. The SMFs show the highest specific compressive modulus of 281.23 kN·m/kg, which is 221.44 times higher than that of PAA foam. Compared to those of traditional polymer or inorganics-based foams, and even to metal foams such as aluminum, SMFs have ultrahigh specific compressive strength and modulus to meet the requirements of load-bearing scenarios.

Owing to the abundant supramolecular interactions in SMFs, it is expected that these foams exhibit easy-to-recycle features which are otherwise hardly possessed by traditional covalently cross-linked foams. The supramolecular cross-linking strategy shows potential for a cradle-to-cradle life cycle, as the foam can be reprocessed into fibers and films by simple water reprocessing.

In summary, we have put forward a SMF with a rare combination of characteristics including lightweight, high mechanical performance, and sustainable recycling ability. LMNPs based on ultrasound-induced acoustic cavitation form oxidized skins with a high specific surface area and a large number of binding sites, which create metal-ligand

coordination with PAA to build supramolecular networks. Meanwhile, nanosizing increases the Gibbs free energy of the system to inhibit the crystallization of LMNPs and allow the foams to be formed at low temperatures. Hence, the obtained foam exhibits high mechanical properties superior to those of Al alloy foam although lightweight, which can withstand the repeated compaction of a car. In addition, benefiting from the rich supramolecular networks, the foam can be reprocessed into fibers and films by simple water reprocessing, showing the potential for a cradle-to-cradle life cycle. We expect this work will provide a new opportunity for the development of extraordinary mechanical properties of lightweight materials towards sustainable applications.

## Methods

### Regents and materials
Acrylic acid (GC, >99%) was purchased from Shanghai Macklin Biochemical Technology Co., Ltd (China). Gallium (Ga, >99.99%) and Indium (In, >99.99%) were purchased from Shanghai Aladdin Biological Technology Co., Ltd (China) and Shanghai Accela Technology (Shanghai) Co., Ltd, respectively. Other chemical reagents were all obtained from Chengdu Kelong Chemical Reagent Company (China). All reagents and ingredients were used without further purification. Deionized water was used in all experiments.

### Preparation of SMFs
First of all, EGaIn was fabricated by heating Ga and In (mass ratio = 3:1) to 120 °C in a nitrogen atmosphere. Different qualities of EGaIn (5%, 10%, and 15% by mass of acrylic acid) were added in 25 g deionized water, and the solution was ultrasounded at 600 W for 10, 20, and 30 min with fixed ultrasonic power of 600 W to make the ultrasonic energy output consistently at 1440 kJ/g during the experiments. Subsequently, acrylic acid was added to the solution and sonicated together at 400 W for 5 min, followed by 5 g of 1% ammonium persulfate was added to the system and sonicated at 200 W for 5 min. An increase in the viscosity of the system can be clearly observed. The high-viscosity liquid was then encapsulated in an airtight container at 60 °C for 8 h. Finally, the SMFs was formation by freeze-drying.

### Recycling process of SMFs
The end-of-life SMFs were immersed in water and the heating to 80 °C for 30 min to dissociate metal-ligand and hydrogen bonds. SMFs became soft and processable during this step. The soft SMFs gel was loaded into the cylinder and the recycled fibers were fabricated by extrusion via high-pressure capillary rheometer (Gottfert RG120, Germany) with a die (length-diameter ratio of 10:1). The recycled fibers were dry to constant weight at room temperature. On the other hand, the films were prepared by hot pressing. The soft SMFs gel was placed in two square metal mold pieces. Films were obtained by hot pressing at 10 MPa and 60 °C for 30 min.

### Characterization
TEM (JEOL JEM-100CX, Japan) was performed to observe the morphology of LMNPs. SEM (JEOL JSM-5900LV, Japan) was used to photograph the pore structure and element distribution of SMFs. DSC mesurements of SMF and PAA foam were performed using TA instruments Q2000 operated at a scanning rate of 5 °C min$^{-1}$ from −70 °C to 40 °C in a nitrogen atmosphere. 2D Raman intensity mapping (20 μm × 20 μm) and the corresponding Raman spectra were recorded on a Raman microscope (Horiba HR Evolution, Japan) with 532 nm laser from 200 to 3400 cm$^{-1}$. AFM measurements were conducted using Bruker icon AFM with an antimony-doped silicon tip (model: RTESPA-300-30) in the PeakForce quantitative nanomechanical modulus mapping mode. The mechanical properties of SMFs were tested by using a universal material testing machine (Instron 5560, USA) in a compression mode with a 1000 N load cell. The mechanical compressive stress and modulus were performed at room temperature and

relative humidity of 40%–50% with a strain speed of 10 mm min$^{-1}$. The software of Abaqus 2020 was used to conduct finite element analysis based on previous literature[53]. If not specified, SMF sample with 10 wt% EGaIn is used for all tests.

### Temperature-dependent FTIR spectroscopy
Temperature-dependent FTIR spectra were acquired using a Fourier transform infrared spectrometer (Nicolet iS50, USA) equipped with a deuterated triacylglycerol sulfate (DTGS) detector. Firstly, PAA/EGaIn was placed on the ZnSe window, and followed by installation of a programmable homemade heating device. After that, the device was heating from 20 °C to 180 °C at the rate of 5 °C min$^{-1}$, FTIR spectra were measured simultaneously in the region of 3800–1000 cm$^{-1}$. Finally, a total of 71 spectra were obtained. The 2D correlation FTIR spectra were processed, calculated, and plotted using 2DCS software. Baseline processing was performed in order to obtain plausible results.

### Broadband dielectric measurements
Novocontrol Concept 50 system with an Alpha impedance analyzer and Quatro Cryosystem temperature control was used to carry to the broadband dielectric measurements. Frequency scans were performed in the range from 10$^{-1}$ to 10$^7$ Hz at the temperatures from −40 to 40 °C, which scanned at intervals of 10 °C. Temperature stability is better than 0.1 K. Analyses of the dielectric spectra were described in Supplementary Note 1 in the Supplementary Information.

## Data availability
All data generated in this study are provided in the article, Supplementary Information, and Source Data file, or from the corresponding author upon request. Source data are provided with this paper.

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

## Acknowledgements

The authors thank the National Natural Science Foundation of China (52373116 and 52173112, received by X.Z.), and the Sichuan Provincial Natural Science Fund for Distinguished Young Scholars (2021JDJQ0017, received by X.Z.) for financial support. They also appreciate Dr. Guiping Yuan from the Analytical and Testing Center and Dr. Lidan Lan from the Polymer Research Institute of Sichuan University for providing TEM tests and mechanical properties testing, respectively.

## Author contributions

Conceptualization, X.Z. and X.Y.; methodology, X.Z. and X.Y.; investigation, X.Y., X.H., X.Q. and Q.G.; writing-original draft, X.Y. and X.Z.; writing – review & editing, X.Y., X.H., X.Q., Q.G. and X.Z.; supervision and project administration, X.Z.

## Competing interests

The authors declare no competing interests.
