## [Peer Review File · Nature Communications]

REVIEWER COMMENTS

Reviewer #1 (Remarks to the Author):

The work by Yang and collaborators deals with the preparation of a metallic foam with high robustness prepared by creating supramolecular networks between poly(acrylic acid) and liquid metal-LM (EGaIn). Nanoscale droplets of LM are stabilized within a polymeric matrix via secondary interactions. The absence of covalent bonds imparts recyclability. Pores are introduced by ice crystals growth, during freezing. After freeze drying, the metallic foam displays a typical characteristic morphology of a cryogel. This is an interesting approach to prepare metallic foams, and the authors have indeed created a highly strong material. However, there are several aspects that need to be clarified, further discussed, and/or properly addressed prior consideration in Nature Communications.

1- While using ultrasonication to convert the bulk LM into nanoscale droplets, the authors have fixed sonication power and time, while varying the LM mass content. This can lead to misleading outcomes, since the amount of energy dissipated to the samples varies possibly masking results coming from formulation. Ultrasonication should be done taking into consideration J/g, therefore the power and/or time should have changed according to the mass fraction of the to-be-sonicated components. In some cases, the authors dissipate half of the energy, for example, when sonicating samples containing 5 or 10% of LM using the same power and time. This needs to be clarified.

2- While referring to supramolecular interactions, there should be more specific discussions. What kind of interactions are taking place? This aspect, which is central to the manuscript, is somewhat overlooked. Metal-polymer interactions are only one taking place, but there are others. On the same note, the term "high-density" interactions is loosely used. How to define this?

3- The authors describe the interactions taking place between metal and polymer as a coordination network. Coordinated networks are usually crystalline and display high long-range order, like MOFs. The interactions taking place during the assembly may be more correctly put at electrostatic interactions. Authors should use XRD or similar analysis to verify the periodicity of the network to call it a coordinated system.

4- The authors affirm that the formation of the core-shell nanostructure (metal core, polymer shell) displayed in Figure 2b modifies the surface tension, but this is rather speculative. The authors need to address the overall assembly mechanism in a more systematic way, possibly using time-resolved experiments. The discussion on the assembly mechanism is not well supported by data.

5- Discussion related to supramolecular structure evidenced by Raman (lines 122-124) is difficult to follow. Figure 2c shows that the metal-polymer complexation is heterogeneous at a given scale. I ask the authors to revisit the discussion and clarify what they mean.

6- The mechanism of "breaking oxide skins" by acrylic acid should be described.

7- The concept of foam is, for some, attributed to those materials where air bubbles were introduced in the system via a foaming agent and mechanical energy (representing a gain in volume). Pore

development of this work is driven by ice crystals growth, so for most this would be considered a cryogel. Please verify the nomenclature.

8- The discussion in line 96-97, and associated SI Fig 5, comparing these foams to PU and AI allow foams is not supported by any data. Nature Communications is a reputable journal and unsupported comparisons can lead to a wide spread of misinformation. PU foams are in the market nowadays, among other reasons, because of its easy processability. In their comparisons the authors infer that their foams are better in terms of processability. This aspect has to be revised.

9- The LM-PAA foams are water sensitive, which is put in the context of possible recyclability in aqueous medium. This is an interesting aspect, but water sensitivity and hygroscopicity is a major drawback for the utilization of these foams in most applications. There should be mechanical data of these foams when subjected to increased humidity.

10- The compression curves clearly indicate differences in pore morphology of the foams depending on the LM content. The authors should add SEM images of their foams, and connect the mechanical results with microstructure, since this is a key aspect of their work: mechanical properties.

11- Overall the authors refer to SI figures in instances where the data is quite relevant to "story" (e.g., line 143). That is an indication that the given figure should, perhaps, be on the main manuscript.

12- The discussion section is merely a brief conclusion. Overall there is no centralized discussion connecting all aspects and arguments raised along the results section. This is highly important since there is a big need to connect process-morphology-properties in this type of research effort, which I missed.

Reviewer #2 (Remarks to the Author):

A supramolecular metallic foam was fabricated. In this SMF, core-shell nanostructured liquid metals connected with high-density supramolecular interactions. The SMFs exhibit high specific strength of 415.45 kN m kg⁻¹ and specific modulus of 281.23 kN m kg⁻¹ which is higher than that of polyacrylic acid. Deep understanding and study are further needed before it can be published. Here are some comments and suggestions.

1. Ligand coordination with PAA method has been reported for many years, e.g. doi.org/10.1002/mame.202200389, doi.org/10.1016/j.colsurfb.2020.111385,

Poly (acrylic acid) coated gold nanoparticles for pH sensing applications (8th European Workshop on Structural Health Monitoring (EWSHM 2016), Bilbao, Spain, 2016), and so on. What's your advantage using liquid metal comparing with metal ions like Zn²⁺, Fe³⁺ or other ligand?

2. From the TEM, the LMNPs seems so big which is in the hundred nanometre scale. Moreover, the size of the LMNPs is hard to control by ultrasonication. So how the uneven metal-ligand can give an enhanced mechanical properties comparing with the uniform ion or molecule ligand? Further study is need.

3. As a liquid, LM has a very small modulus. When the LMNPs used as cross-linking point, how could a flexible cross-linking point lead to an improved mechanical strength of the whole material?

4. Ga can turn to Ga ions in the condition of acid. So, the real cross-linking point may be Ga³⁺ but not the LMNPs. Related experiments should be given.

5. The author claimed: LMNPs were uniformly distributed in the foam skeleton without agglomeration (Supplementary Figure 4-5). In Figure 4-5, it is hard to see the Ga and In, and the scale bar is 200 μm which is too big to claim a uniform distribution.

6. The author claimed: phase change won't occur in LMs during the freeze-drying process (Supplementary Figure 1). However, there is a small peak around -25°C. Take into consideration of the small content of LMNPs, it can not be ignored. And the DSC of higher contents of LMNPs should be given.

Point by Point Response to Reviewer Comments

To the reviewer 1:

The work by Yang and collaborators deals with the preparation of a metallic foam with high robustness prepared by creating supramolecular networks between poly(acrylic acid) and liquid metal-LM (EGaIn). Nanoscale droplets of LM are stabilized within a polymeric matrix via secondary interactions. The absence of covalent bonds imparts recyclability. Pores are introduced by ice crystals growth, during freezing. After freeze drying, the metallic foam displays a typical characteristic morphology of a cryogel. This is an interesting approach to prepare metallic foams, and the authors have indeed created a highly strong material. However, there are several aspects that need to be clarified, further discussed, and/or properly addressed prior consideration in Nature Communications.

We acknowledge your comments and suggestions. We revised our manuscript in accordance with your instructive guidance and we feel that the revised manuscript is a significant improvement on the original one. Hopefully we have addressed your concerns. Thank you very much.

- 1) While using ultrasonication to convert the bulk LM into nanoscale droplets, the authors have fixed sonication power and time, while varying the LM mass content. This can lead to misleading outcomes, since the amount of energy dissipated to the samples varies possibly masking results coming from formulation. Ultrasonication should be done taking into consideration J/g, therefore the power and/or time should have changed according to the mass fraction of the to-be-sonicated components. In some cases, the authors dissipate half of the energy, for example, when sonicating samples containing 5 or 10% of LM using the same power and time. This needs to be clarified.

✓ According to the reviewer, we have adjusted the ultrasonic time to be 10, 20 and 30 min with fixed ultrasonic power of 600 W, which make the ultrasonic energy output consistently at 1440 kJ/g during the experiments. Compressive stress-strain curves of SMFs based on the above experiments are shown in Figure

3b-c. The experimental procedures of the Revised Manuscript have been modified correspondingly (Page 18, Line 337-339). Thank you very much for your careful work.

- 2) While referring to supramolecular interactions, there should be more specific discussions. What kind of interactions are taking place? This aspect, which is central to the manuscript, is somewhat overlooked. Metal-polymer interactions are only one taking place, but there are others. On the same note, the term “high-density” interactions is loosely used. How to define this?

✓ Based on the reviewer’s concern, supramolecular interactions have been discussed more specifically in the Revised Manuscript (Page 1, Line 15-16). Metal-ligand coordination formed between PAA and LMNPs, hydrogen bonding interactions formed by PAA itself are taking place simultaneously in SMFs. To explain the high-density interactions, 2D Raman intensity surface mapping of 1719 cm^{-1} on the surface of SMF and PAA foam has been conducted, which intuitively illustrate the distribution of gallium-carboxyl complexation. The high peak intensity proves high-density metal-ligand interactions of SMF compared to PAA foam (Supplementary Figure 11). The relevant discussion has been added to the Revised Manuscript (Page 6, Line 132-133; Page 7, Line 136-140). Thanks for your careful works.

- 3) The authors describe the interactions taking place between metal and polymer as a coordination network. Coordinated networks are usually crystalline and display high long-range order, like MOFs. The interactions taking place during the assembly may be more correctly put at electrostatic interactions. Authors should use XRD or similar analysis to verify the periodicity of the network to call it a coordinated system.

✓ According to the reviewer, XRD was used to analyze the periodicity of the network shown in Supplementary Figure 6. The SMF shows amorphous state, which is not the cyclical structure. This is because that coordination bonds can be tuned in a rather broad range by changing the type of acid or base. Some polymer networks based on strong coordination bonds can form long-range ordered structures, such as MOFs. The coordination bonds between Ga (III) and carboxylate have higher kinetic activity for dissociation and recombination. Such coordination bonds make it difficult to form periodic structures such as crystals, which is also confirmed by previous literatures^{1,2}. To make the description clearer, we have discussed it more in the Revised Manuscript (Page 5, Line 105-108) to help readers understand it. Thank you very much.

- 4) The authors affirm that the formation of the core-shell nanostructure (metal core, polymer shell) displayed in Figure 2b modifies the surface tension, but this is rather speculative. The authors need to address the overall assembly mechanism in a more systematic way, possibly using time-resolved experiments. The discussion on the assembly mechanism is not well supported by data.

√ According to the reviewer's suggestion, time-resolved zeta potential has been used to analyze the assembly mechanism of core-shell nanostructure. The change in zeta potential with time proves that acrylic acid monomer progressively altered the surface charges during ultrasonication. The repulsive force between nanoparticles increased due to the enhanced negative charge carried. The relevant discussion has been added to the Revised Manuscript (Page 4-5, Line 85-90). We are very grateful to reviewer's pertinent and helpful suggestion.

- 5) Discussion related to supramolecular structure evidenced by Raman (lines 122-124) is difficult to follow. Figure 2c shows that the metal-polymer complexation is heterogenous at a given scale. I ask the authors to revisit the discussion and clarify what do they mean.

✓ The appearance of Raman spectral peak at 1719 cm^{-1} proves the coordination interactions between Ga (III) and carboxyl groups. Furthermore, 2D Raman mapping was developed by measuring the intensity of 1719 cm^{-1} on the surface ($20\text{ }\mu\text{m} * 20\text{ }\mu\text{m}$) of SMF. The strong color contrast indicates the different intensity for metal-ligand coordination at different locations of SMF. No obvious metal-ligand coordination peaks appear in 2D Raman mapping of PAA foam. This is attributed to the fact that the complexation is related to the distribution of LMNPs. The relevant discussion has been added to the Revised Manuscript (Page 7, Line 136-140). Hopefully we have addressed your concerns.

- 6) The mechanism of “breaking oxide skins” by acrylic acid should be described.

✓ Thank you very much for your suggestions. The mechanism of “breaking oxide skins” by acrylic acid have been described in the Revised Manuscript (Page 4, Line 77-80). The specific mechanism is that liquid metals of EGaIn can develop a near-instant gallium oxide layer consisted primarily of Ga₂O₃ when exposed to oxygen. Hydrogen ions ionized of acrylic acid can react with Ga₂O₃ to form Ga³⁺ and water to break the oxide skins, which has also been reported previously in the literatures^{3,4}.

- 7) The concept of foam is, for some, attributed to those materials where air bubbles was introduced in the system via a foaming agent and mechanical energy (representing a gain in volume). Pore development in this work is driven by ice crystals growth, so for most this would be considered a cryogel. Please verify the nomenclature.

✓ Thank you very much for your suggestions. Previous literatures has collectively referred to macroporous materials fabricated by the methods (e.g., introducing bubbles⁵, vapor deposition⁶, 3D printing^{7,8} and freeze casting⁹⁻¹¹) as foams. In order not to mislead the readers, the definition of our foams has been given in the Introduction of Revised Manuscript (Page 2, Line 35-36) according to the reviewer’s suggestion.

- 8) The discussion in line 96-97, and associated SI Fig 5, comparing these foams to PU and Al alloy foams is not supported by any data. Nature Communications is a reputable journal and unsupported comparisons can lead to a wide spread of misinformation. PU foams are in the market nowadays, among other reasons, because its easy processability. In their comparisons the authors infer that their foams are better in terms of processability. This aspect has to be revised.

✓ To clarify the comparative basis of different foams, we have compiled information on the variety of mechanical durability, recyclability, processability, density and compressibility of SMF, PU foam and Al alloy foam. We have listed the results in Supplementary Table 1. The corresponding radar chart has been redrawn in Supplementary Figure 10. Hopefully we have addressed your concerns.

Supplementary Table 1. Comparison about mechanical durability, recyclability, processability, density and compressibility of SMF, PU foam and Al alloy foam.

Type of foams	SMF	PU foam	Al alloy foam
Mechanical durability	Specific Modulus: 281.23 kN m kg ⁻¹	Specific Modulus: ~46-68 kN m kg ⁻¹	Specific Modulus: ~200 kN m kg ⁻¹
Recyclability	Water reprocessing at 80 °C	Crosslinking structure. Hard to recycle.	Melting of scrap > 600 °C
Processability	Freeze-drying	Foaming and curing at 150 °C	Foaming at ~ 800 °C
Density	0.20-0.32 g cm ⁻³	~ 0.25 g cm ⁻³	0.43-0.8 g cm ⁻³
Compressibility	Compressive strain > 90%	Compressive strain > 90%	Brittle cracking at ~10% strain
Ref.	This work	[12,13]	[14,15]

- 9) The LM-PAA foams are water sensitive, which is put in the context of possible recyclability in aqueous medium. This is an interesting aspect, but water sensitivity and hygroscopicity is a major drawback for the utilization of these foams in most applications. There should be mechanical data of these foams

when subjected to increased humidity.

✓ The compressive mechanical properties of SMF (10 wt% of EGaIn) with increased humidity has been conducted based on the reviewer's comments. The mechanical compressive stress of original sample was performed at relative humidity of 40%-50% which meets most scenarios for practical use. And the mechanical properties of SMF were decreased with the increased humidity as shown in Supplementary Figure 18. Compressive modulus of SMF at humidity of 90% is still about 5 times higher than PU foam. The relevant discussion has been added to the Revised Manuscript (Page 11, Line 222-224). In our subsequent research, we will endeavor to improve the moisture resistance of foams without compromising the recyclability of the materials, for example, by structural design such as covalent-noncovalent interpenetrating network. Hopefully we have addressed your concerns.

10) The compression curves clearly indicate differences in pore morphology of the foams depending on the LM content. The authors should add SEM images of their foams, and connect the mechanical results with microstructure, since this is a key aspect of their work: mechanical properties.

✓ According to the reviewer, we have added SEM images of SMFs under different compressive strain. When under small strain compression (5%), the pore morphology of foams didn't change noticeably. As the strain increases to 50%,

partial squeezing of the porous structures was found. Ultimately, as SMFs were further compressed (85%), most of the holes were compressed and the high strength provided by the skeleton. We have combined stress-strain curves with the changes in the pore morphology to describe the mechanical properties of the foams in the Revised Manuscript (Page 10, Line 201-210). We are very grateful to your pertinent and helpful suggestion.

11) Overall the authors refer to SI figures in instances where the data is quite relevant to “story” (e.g., line 143). That is an indication that the given figure should, perhaps, be on the main manuscript.

✓ Based on the reviewer’s concern, PCMW2D synchronous spectra has been given in the main manuscript (Figure 2f) of Revised Manuscript. Furthermore, Figure 3 has also been realigned. Thank you very much.

12) The discussion section is merely a brief conclusion. Overall there is no centralized discussion connecting all aspects and arguments raised along the results section. This is highly important since there is a big need to connect process-morphology-properties in this type of research effort, which I missed.

✓ Based on the reviewer's comments, the connection of process-morphology-properties has been centralized discussed in the discussion section of Revised Manuscript (Page 12-13, Line 259-278). We are very grateful to reviewer's pertinent and helpful suggestion.

To the reviewer 2:

A supramolecular metallic foam was fabricated. In this SMF, core-shell nanostructured liquid metals connected with high-density supramolecular interactions. The SMFs exhibit high specific strength of 415.45 kN m kg⁻¹ and specific modulus of 281.23 kN m kg⁻¹ which is higher than that of polyacrylic acid. Deep understanding and study are further needed before it can be published. Here

are some comments and suggestions.

We acknowledge your positive comments and suggestions. We have revised our manuscript in accordance with your instructive guidance. Hopefully, we have addressed your concerns. Thank you very much.

- 1) Ligand coordination with PAA method has been reported for many years, e.g. doi.org/10.1002/mame.202200389, doi.org/10.1016/j.colsurfb.2020.111385, Poly (acrylic acid) coated gold nanoparticles for pH sensing applications (8th European Workshop on Structural Health Monitoring (EWSHM 2016), Bilbao, Spain, 2016), and so on. What's your advantage using liquid metal comparing with metal ions like Zn^{2+} , Fe^{3+} or other ligand?

✓ The introduction of LMNPs to form metal-ligand coordination with PAA has following advantages. Nanoscale LMNPs exhibit high specific surface area as physical cross-linking sites, whose oxidized skins can form abundant metal-ligand coordination to provide high skeleton strength. At the same time, flowability of LMNPs helps transfer and dissipate energy under stress, thus avoiding damage of foams due to stress concentrations. Furthermore, LMNPs also contribute to the foam shaping process. Nanosizing increases Gibbs free energy of the system thereby inhibiting crystallization of LMs during freeze-drying, which aids in the load transfer of the process and prevents hole collapse. Hopefully we have addressed your concerns.

- 2) From the TEM, the LMNPs seems so big which is in the hundred nanometre scale. Moreover, the size of the LMNPs is hard to control by ultrasonication. So how the uneven metal-ligand can give an enhanced mechanical properties comparing with the uniform ion or molecule ligand? Further study is need.

✓ According to the reviewer, we have investigated the mechanical properties of foam with homogeneous metal-ligand bonding networks by introducing $GaCl_3$ into PAA matrix. The PAA/ Ga^{3+} foam shows brittle destruction of porous structure leading to stress loss during compression (shown in the red dashed box of Supplementary Figure 17). Different from the uniform metal-ligand cross-linking,

flowability of nano-scaled LMNPs helps transfer and dissipate energy through deformation at the same time, which contributes significantly to the enhancement of the mechanical toughness of the metallic foams. Relevant descriptions can be found in the Revised Manuscript (Page 10, Line 209-210). Thank you for your helpful suggestion.

- 3) As a liquid, LM has a very small modulus. When the LMNPs used as cross-linking point, how could a flexible cross-linking point lead a improved mechanical strength of the whole material?

✓ Recent work has demonstrated that high modulus of materials can be achieved by rich coordination networks². In our work, nanoscale LMNPs exhibit high specific surface area as physical cross-linking sites, whose oxidized skins can form abundant metal-ligand coordination to provide high skeleton strength. At the same time, LMNPs can deform to dissipate energy when subjected to external forces, simultaneous achieving high modulus and high toughness of foams. Hopefully we have addressed your concerns.

- 4) The Ga can turn to Ga ions in the condition of acid. So, the real cross-linking point may be Ga³⁺ but not the LMNPs. Related experiments should be given.

✓ Hydrogen ions ionized of acrylic acid can react with Ga₂O₃ to form Ga³⁺ and water to break the oxide skins^{3,4}. According to the reviewer, XPS has been used to analyze the valence states of Ga element. As shown in Supplementary Figure 3,

abundant Ga^{3+} (49.3 atom% of Ga element) have been found in XPS curves, attributed to the Ga_2O_3 layer on the surface of LMNPs and the formation of Ga^{3+} by reaction with acid. However, Ga_2O_3 and free Ga^{3+} cannot be distinguished in XPS. Furthermore, high-resolution TEM image of the core-shell structure was obtained to discover the oxide layer between LMNP and organic shell. The change in zeta potential with time also proved that absorption of acrylic acid on surfaces of LMNPs and thus modulation of surface charge, demonstrating the interactions between LMNPs and acrylic acid. Hence, both the oxidized skins of LMNPs and the free Ga^{3+} in the vicinity will form metal-ligand coordination with carboxyl groups on the macromolecular chains simultaneously thus forming nanostructured supramolecular networks. Relevant descriptions can be found in the Revised Manuscript (Page 4-5, Line 85-93). Thank you very much.

5) The author claimed: LMNPs were uniformly distributed in the foam skeleton without agglomeration (Supplementary Figure 4-5). In Figure 4-5, it is hard to see the Ga and In, and the scale bar is 200um which is too big to claim a uniform distribution.

✓ According to reviewer's comments, the SEM images of foam skeleton with greater magnification have been added in the Supplementary Information of the Revised Manuscript. As shown in Supplementary Figure 8, the LMNPs can be observed on the surface and cross section of foam skeleton, which are uniformly distributed without obvious agglomeration. This result can also be affirmed by the EDS mapping of Ga and In elements.

6) The author claimed: phase change won't occur in LMs during the freeze-drying process (Supplementary Figure 1). However, there is a small peak around -25°C . Take into consideration of the small content LMNPs, it can not be ignored. And the DSC of higher contents of LMNPs should be given.

✓ Based on the reviewer's comments, DSC curves of higher content of LMNPs has been given in the Revised Manuscript. As shown in Supplementary Figure 4, when the content of LMNPs was increased, there are no obvious melting peaks to be found. Due to the size dependence of the phase transition temperature of LMNPs. A small peak at -26°C is attributed to the phase transition peak of a very small number of large-sized LMNPs. The Revised Manuscript have been modified correspondingly (Page 5, Line 102-103). Thank you very much for your careful work.

References

1. Yang, Q. *et al.* Strong and Tough Antifreezing Hydrogel Sensor via the Synergy of Coordination and Hydrogen Bonds. *ACS Appl. Mater. Interfaces* **15**, 51684–51693 (2023).
2. Lai, J.-C. *et al.* A rigid and healable polymer cross-linked by weak but abundant Zn(II)-carboxylate interactions. *Nat. Commun.* **9**, 2725 (2018).
3. Liu, S., Shah, D. S. & Kramer-Bottiglio, R. Highly stretchable multilayer electronic circuits using biphasic gallium-indium. *Nat. Mater.* **20**, 851–858 (2021).
4. Daeneke, T. *et al.* Liquid metals: fundamentals and applications in chemistry. *Chem. Soc. Rev.* **47**, 4073–4111 (2018).
5. Peyrton, J. & Avérous, L. Structure-properties relationships of cellular materials from biobased polyurethane foams. *Mater. Sci. Eng. R Reports* **145**, 100608 (2021).
6. Wang, S. *et al.* Novel flower-like graphene foam directly grown on a nickel template by chemical vapor deposition. *Carbon N. Y.* **120**, 103–110 (2017).
7. Chen, Q., Cao, P.-F. & Advincula, R. C. Mechanically Robust, Ultraelastic Hierarchical Foam with Tunable Properties via 3D Printing. *Adv. Funct. Mater.* **28**, 1800631 (2018).
8. Krisnadi, F. *et al.* Printable Liquid Metal Foams That Grow When Watered. *Adv. Mater.* **n/a**, 2308862 (2024).
9. Cai, S. *et al.* Ultralong Organic Phosphorescent Foams with High Mechanical Strength. *J. Am. Chem. Soc.* **143**, 16256–16263 (2021).
10. Di, A., Schiele, C., Hadi, S. E. & Bergström, L. Thermally Insulating and Moisture-Resilient Foams Based on Upcycled Aramid Nanofibers and Nanocellulose. *Adv. Mater.* **35**, 2305195 (2023).

11. Zhao, S., Malfait, W. J., Guerrero-Alburquerque, N., Koebel, M. M. & Nyström, G. Biopolymer Aerogels and Foams: Chemistry, Properties, and Applications. *Angew. Chemie Int. Ed.* **57**, 7580–7608 (2018).
12. Stanzione, M. *et al.* Tuning of polyurethane foam mechanical and thermal properties using ball-milled cellulose. *Carbohydr. Polym.* **231**, 115772 (2020).
13. Li, H., Sinha, T. K., Oh, J. S. & Kim, J. K. Soft and Flexible Bilayer Thermoplastic Polyurethane Foam for Development of Bioinspired Artificial Skin. *ACS Appl. Mater. Interfaces* **10**, 14008–14016 (2018).
14. Lin, Y., Zhang, Q., Ma, X. & Wu, G. Mechanical behavior of pure Al and Al–Mg syntactic foam composites containing glass cenospheres. *Compos. Part A Appl. Sci. Manuf.* **87**, 194–202 (2016).
15. Aldoshan, A. & Khanna, S. Effect of relative density on the dynamic compressive behavior of carbon nanotube reinforced aluminum foam. *Mater. Sci. Eng. A* **689**, 17–24 (2017).

REVIEWER COMMENTS

Reviewer #1 (Remarks to the Author):

I am pleased to say that the authors have amended all comments and issues raised during the first review round. Their work is now suitable for publication in Nature Communications. Great work, congratulations.

Reviewer #2 (Remarks to the Author):

The authors still have not given the scientific proof how LMNPs can increase the strength.

The author claimed in the rebuttal: "Oxidized skins can form abundant metal-ligand coordination to provide high skeleton strength, and flowability of LMNPs helps transfer and dissipate energy under stress, thus avoiding damage of foams due to stress concentrations." All of the discussions are based on language description, without scientific data or control experiments.

1. Adding Ga₂O₃ nano particles or Ga³⁺ separately without LMNPs can show the real role of LMNPs.
2. Finite element analysis can also show the possibility how soft crosslinking points can change the strength of the system.

Point by Point Response to Reviewer Comments

To the reviewer 1:

I am pleased to say that the authors have amended all comments and issues raised during the first review round. Their work is now suitable for publication in Nature Communications. Great work, congratulations.

We are grateful for the reviewer's positive recommendation in acceptance of our manuscript and insightful comments that help greatly improve the quality of the manuscript.

To the reviewer 2:

The authors still have not given the scientific proof how LMNPs can increase the strength.

The author claimed in the rebuttal: "Oxidized skins can form abundant metal-ligand coordination to provide high skeleton strength, and flowability of LMNPs helps transfer and dissipate energy under stress, thus avoiding damage of foams due to stress concentrations." All of the discussions are based on language description, without scientific data or control experiments.

We acknowledge your comments and suggestions. Control experiments and finite element analysis have been added in the Revised Manuscript to prove the role of LMNPs. We have revised our manuscript in accordance with your instructive guidance. Hopefully, we have addressed your concerns. Thank you very much.

- 1) Adding Ga₂O₃ nano particles or Ga³⁺ separately without LMNPs can show the real role of LMNPs.

√ According to the reviewer, we have investigated the mechanical properties of foam adding Ga³⁺ without LMNPs. The PAA/Ga³⁺ foam shows brittle destruction of porous structure leading to stress loss during compression (shown in the red dashed box of Supplementary Figure 17). Different from the cross-linking of Ga³⁺, flowability of nano-scaled LMNPs helps transfer and dissipate energy through

deformation at the same time to improve the strength and toughness of foams, which have also been affirmed by finite element analysis.

2) Finite element analysis can also show the possibility how soft crosslinking points can change the strength of the system.

√ Finite element analysis has been conducted to study how soft crosslinking points change the strength of the system based on reviewer's comments. As shown in Supplementary Figure 18-19, large deformation of LMNPs reduces the load at the interface between LMNPs and polymer matrix. On the contrary, higher stress at the surface between the polymer matrix and rigid nanoparticles can easily damage or destroy the materials. This result suggests that flowability and large deformation of LMNPs can transfer and dissipate energy under stress, thus avoiding damage of foams due to stress concentrations. The relevant discussion has been added to the Revised Manuscript (Page 10, Line 210-216). We are very grateful to reviewer's pertinent and helpful suggestion.

REVIEWERS' COMMENTS

Reviewer #2 (Remarks to the Author):

The author's reply has given full and reasonable response for all the comments. It can be accepted in present version.